# Epicatechin-Loaded Nanocapsules: Development, Physicochemical Characterization, and NLRP3 Inflammasome-Targeting Anti-Inflammatory Activity

**DOI:** 10.3390/biology14111520

**Published:** 2025-10-30

**Authors:** Carolina Bordin Davidson, Éricles Forrati Machado, Amanda Kolinski Machado, Diulie Valente de Souza, Lauren Pappis, Giovana Kolinski Cossettin Bonazza, Djenifer Letícia Ulrich Bick, Taíse Regina Schuster Montagner, André Gündel, Ivana Zanella da Silva, Aline Ferreira Ourique, Alencar Kolinski Machado

**Affiliations:** 1Graduate Program in Nanosciences, Franciscan University, Santa Maria 97010-030, Brazil; carolina.davidson@ufn.edu.br (C.B.D.); ericles.forrati@ufn.edu.br (É.F.M.); diulie.souza@ufn.edu.br (D.V.d.S.); giovana.bonazza@ufn.edu.br (G.K.C.B.); djenifer.ulrich@ufn.edu.br (D.L.U.B.); ivanazanella@ufn.edu.br (I.Z.d.S.); aline.ourique@ufn.edu.br (A.F.O.); 2Laboratory of Cell Culture and Bioactive Effects, Franciscan University, Santa Maria 97010-030, Brazil; amanda.kolinski@ufn.edu.br (A.K.M.); taise.schuster@ufn.edu.br (T.R.S.M.); 3Nanostructured Systems Research Laboratory, Franciscan University, Santa Maria 97010-030, Brazil; 4Undergraduate Course in Pharmacy, University of Western Santa Catarina, Joaçaba 89600-000, Brazil; lauren.pappis@unoesc.edu.br; 5Laboratory of Atomic Force Microscopy, Federal University of Pampa, Bagé 96413-170, Brazil; andregundel@unipampa.edu.br

**Keywords:** bioactive molecule, epicatechin, flavonoids, inflammation, nanocapsule, nanotechnology

## Abstract

This experimental study reports the development and physicochemical characterization of epicatechin-containing nanocapsules, as well as their in vitro bioactive effects in a monolayer cell culture model. To this end, the epicatechin-containing nanocapsules were evaluated for their safety profile and anti-inflammatory efficacy in macrophages via modulation of the NLRP3 inflammasome.

## 1. Introduction

Inflammation is a physiological process that is an essential component of the body’s defense mechanisms; and is mediated by the coordinated activation of immune cells and the release of inflammatory mediators in response to pathogens, injuries, or other harmful stimuli [1]. However, when this inflammation response becomes exacerbated or chronic, it can contribute to the genesis and progression of various diseases, such as metabolic, cardiovascular, or neurodegenerative disorders, and cancers [2]. The NLRP3 inflammasome is an intracellular multiprotein complex that triggers an inflammatory response in the presence of endogenous and exogenous danger signals, culminating in activation of caspase-1 and subsequent secretion of pro-inflammatory cytokines, such as IL-1β and IL-18 [3]. Dysregulated NLRP3 activation has been implicated in a wide range of chronic inflammatory disorders, making it a strategic target for the development of new therapies. Because currently available anti-inflammatory drugs have limitations, including low specificity and significant side effects [4,5], identification and development of new agents capable of selectively modulating NLRP3 could enable more effective management of complex inflammatory conditions.

Flavonoids are natural polyphenolic molecules that are widely distributed in fruits, vegetables, and medicinal plants, and are known for their biological activity, including their antioxidant, anti-inflammatory, and therapeutic potential [6,7,8]. Epicatechin is a catechin-type flavonoid found in green tea and açaí which has been reported to modulate oxidative stress and inflammatory pathways, including mechanisms involving the NLRP3 inflammasome [9,10,11].

Despite its biological potential, several physicochemical properties of epicatechin have hindered its clinical applications, including its low aqueous solubility, poor bioavailability, and high susceptibility to chemical and enzymatic degradation in biological or environmental conditions [12]. Factors such as pH, temperature, light, and the presence of oxygen can significantly affect the stability of epicatechin, leading to a rapid loss of activity and reduced therapeutic efficacy [13,14,15]. Consequently, strategies to improve the stability, solubility, and delivery of epicatechin are essential to expanding its pharmacological potential [16].

Nanotechnology offers innovative solutions to these challenges, by enabling encapsulation of bioactive molecules in nanocarriers [17,18]. Polymeric nanocapsules have been designed that are capable of protecting sensitive compounds, enhancing solubility, improving absorption, and promoting targeted compound delivery [19,20]. Eudragit L-100 is an anionic copolymer that is widely employed in nanotechnology for drug delivery, because of its pH-sensitive properties and biocompatibility [21,22]. In fact, Eudragit L-100 has been shown to provide protection for encapsulated compounds under acidic gastric conditions, and enables compound release in an intestinal environment, which is particularly advantageous for overcoming the low solubility, instability, and poor bioavailability that typically limits the therapeutic potential of flavonoids. Previous studies have shown that nanocapsules prepared with Eudragit L-100 can enhance the stability and biological performance of poorly soluble compounds, such as curcumin [23].

This study aims to develop, for the first time, NC-ECs using Eudragit^®^ L-100 via interfacial deposition of a preformed polymer. The study consists of two phases: (1) formulation, physicochemical characterization, and evaluation of the thermal stability of the NC-ECs; and (2) preliminary assessment of its safety profile using a VERO cell line, and evaluation of the in vitro anti-inflammatory efficacy of the NC-ECs, specifically their ability to modulate the NLRP3 inflammasome in LPS + nigericin-stimulated THP-1-derived macrophages.

## 2. Materials and Methods

The present study has two main objectives: (1) the formulation, comprehensive physicochemical characterization, and stability assessment of an NC-EC; and (2) preliminary evaluation of its cytocompatibility and in vitro anti-inflammatory efficacy, specifically with respect to targeting NLRP3 inflammasome modulation (Figure 1).

### 2.1. Part I

#### 2.1.1. Nanocapsule Preparation

Nanocapsules were prepared via interfacial deposition of a preformed polymer, following methodology adapted from Coradini et al. [24] and Jaguezeski et al. [23]. The organic phase contained the Eudragit^®^ L-100 polymer (1%), medium chain triglyceride (1.65%), epicatechin (0.025%), and sorbitan monostearate (0.4%), dissolved in ethyl alcohol (54 mL) for 15 min at 40 °C. The organic phase was poured into an aqueous phase containing polysorbate 80 (0.75%) and ultrapure water (108 mL). After stirring for 15 min, the solvent was evaporated under reduced pressure at 40 °C with the aid of a rotary evaporator, until a final volume of 20 mL was obtained. The final concentration of NC-ECs was 0.25 mg/mL. Blank nanoparticles (NC-Bs) were prepared using the same methodology, but without the addition of epicatechin.

#### 2.1.2. Nanocapsule Characterization

Nanocapsules prepared as above were characterized according to average particle size, polydispersity index (PDI), zeta potential, pH, bioactive content, encapsulation efficiency (EE), and morphology.

Average particle size and PDI were measured by dynamic light scattering (DLS) using a Zetasizer^®^ (Zetasizer^®^ Nano-ZS model ZEN 3600, Malvern Instruments^®^, Malvern, UK), equipped with a laser source at a wavelength of 532 nm and a measurement angle of 173°. For DLS measurements, samples were diluted 500× (*v*/*v*) in ultrapure water. The results are expressed in nanometers (nm) for the diameter and analyzed based on the average of three replicates. Determination of zeta potential was performed in triplicate using an electrophoretic mobility technique available in the Zetasizer^®^ (Zetasizer^®^ Nano-ZS model ZEN 3600, Malvern Instruments^®^), with a laser source set at a wavelength of 532 nm and a measurement angle of 173°. The results are expressed in millivolts (mV). For electrophoretic measurements, samples were diluted 500× (*v*/*v*) in NaCl 10 mM. The pH was determined using a potentiometer (Denver Instrument^®^, Arvada, CO, USA) previously calibrated with standard pH 4.0 and 7.0 buffer solutions. The results were expressed as an average of three pH readings.

The amount of bioactive molecule in the samples was quantified via reversed-phase HPLC (RP-HPLC) on a Prominence^®^ chromatograph (Shimadzu^®^, Kyoto, Japan), equipped with a pump (model LC-20AT), and a PDA ultraviolet detector (model SPD-M20A) with variable wavelength UV/VIS. Chromatography was conducted using an RP-C18 chromatography column (250 × 4.60 mm, 5 mm, Phenomenex^®^, Torrance, CA, USA) and a pre-column 4 × 3 mm, 5 mm (Phenomenex^®^, Torrance, CA, USA). Chromatographic conditions were selected based on a study developed by Klein, Longhini, and de Mello [25], and results from the method were linear between concentrations of 10 and 70 μg/mL (*y* = 134,410x − 16,925, *R*^2^ = 0.9998).

The mobile phase was composed of phase A, water with 0.1% trifluoroacetic acid (TFA), and phase B, methanol:acetonitrile (25:75 *v*/*v*) at a 90:10 *v*/*v* ratio (A:B). Chromatography was conducted with the flow rate of the mobile phase set at 1 mL/min, an injection volume of 20 µL, isocratic elution, and an oven temperature of 30 °C.

To assay the bioactivity of the preparation, 1.6 mL of the formulation was transferred to a 10 mL flask and the sample was adjusted to a final volume of 10 mL by dilution with mobile phase. Samples were then centrifuged for 30 min at 7000× *g*, filtered through a 0.45 µm membrane, and injected into the HPLC to quantify epicatechin content. The EE was determined via an ultrafiltration–centrifugation technique using ultrafiltration devices (Microcon–Millipore 10 kDa, Burlington, MA, USA). The ultrafiltrate was obtained by centrifugation of 500 μL of nanocapsules containing the bioactive ingredient for 30 min at 7000× *g* [26]. The EE was calculated as the difference between the total amount of bioactive ingredient in the formulation (content) and the amount of free bioactive ingredient (ultrafiltrate). Determination of the amount of free bioactive ingredient present in the ultrafiltrate was evaluated by HPLC, according to the method described previously.

Morphological analysis of the nanocapsules was performed by atomic force microscopy, following the methodology described by Gündel et al. [27].

#### 2.1.3. Centrifugation Test

A centrifugation test was used to verify the preliminary stability of the prepared NC-ECs, according to the Cosmetic Product Stability Guide/ANVISA [28]. This test produces an increase in the force of gravity, increasing particle mobility and enabling prediction of possible instability. For this test, samples were centrifuged at 1300× *g* for 30 min at room temperature (25 ± 2 °C). The formulations were then macroscopically evaluated for phase separation and images were photographed.

#### 2.1.4. Thermal Stability of Nanocapsules

To evaluate the stability of the prepared NC-ECs, three groups (*n* = 3) were produced and divided into three temperature conditions: room temperature (25 ± 2 °C), climate chamber (40 ± 2 °C), and refrigeration (5 ± 2 °C). Samples were analyzed for average size, zeta potential, pH, and PDI at time 0 (the day of preparation) and at 7, 15, 30, 45, and 60 days post production. In addition, the odor and appearance of the suspension was evaluated. All analyses were performed in triplicate and the results are expressed as the mean ± standard deviation, with statistical analyses performed in comparison with formulations made immediately after preparation [27].

### 2.2. Part II

#### 2.2.1. Protein Corona Assay

The NC-ECs were diluted at a concentration of 50 µg/mL in the respective culture media, RPMI 1640 and complete DMEM, both supplemented with 10% (*v*/*v*) fetal bovine serum (FBS), 1% (*v*/*v*) 100 U/mL penicillin, and 100 mg/mL of streptomycin. Subsequently, they were incubated in a CO_2_ incubator at 37 °C and analyzed at times (t) 0, 24, 48, and 72 h, where t = 0 was used as a baseline. The physicochemical parameters of the prepared NC-ECs, such as average particle diameter, PDI, zeta potential, and pH, were measured at each time point and compared to the baseline at t = 0.

#### 2.2.2. In Vitro Safety Profile of the Prepared Nanocapsules

Kidney epithelial cells (VERO cell line, ATCC^®^ CCL-81™) were obtained from the Rio de Janeiro Cell Bank (BCRJ, 0245, Rio de Janeiro, RJ, Brazil) and cultured using Dulbecco’s modified Eagle medium (DMEM) supplemented with 10% FBS and 1% (*v*/*v*) penicillin/streptomycin. For investigation of the in vitro safety profile of NC-ECs and NC-Bs, VERO cells were seeded at 1.5 × 10^5^ cells/mL in 96-well plates. The cells were exposed to increasing concentrations of NC-ECs or NC-Bs (0.01–50 µg/mL) for 24, 48, and 72 h. After the incubation period, the effects of NC-ECs and NC-Bs on cellular viability, cell proliferation indexes, nitric oxide (NO) production, reactive oxygen species (ROS) levels, and release of double-stranded DNA (dsDNA), were evaluated.

#### 2.2.3. In Vitro Anti-Inflammatory Effects of NC-ECs

The potential anti-inflammatory activity of NC-ECs was evaluated in vitro, by monitoring NLRP3 inflammasome modulation. For this purpose, monocytes from a THP-1 cell line (ATCC TIB-202^TM^, purchased from BCRJ) were cultured in RPMI 1640 medium supplemented with 10 mM HEPES buffer, 10% FBS and 1% (*v*/*v*) penicillin/streptomycin. The cells were incubated in 5% CO_2_ at 37 °C. First, monocytes were induced to differentiate into macrophages using 25 ng/mL phorbol 12-myristate 13-acetate (PMA) [29]. THP-1-derived macrophages were then activated for inflammation via the NLRP3 inflammasome pathway with LPS + nigericin, following a protocol previously described in Davidson et al. [10]. The LPS + nigericin-activated cells were treated with increasing concentrations of NC-ECs ranging from 0.01 to 50 μg/mL. MCC950, a known synthetic NLRP3 inhibitor, was used as an inhibition control. Finally, the LPS + nigericin-activated macrophages that were treated with NC-ECs were analyzed for cell viability, NO levels, total ROS levels, and dsDNA levels in the extracellular medium. Gene expression of cytokines, NLRP3, and caspase-1 was measured in cells treated with the optimal NC-EC concentration, which resulted in the greatest reversal of LPS + nigericin-induced inflammation.

#### 2.2.4. Experimental Analysis

Cell viability and proliferation (MTT) assays, nitric oxide (NO) quantification, intracellular reactive oxygen species (ROS) measurements, extracellular DNA (dsDNA) release, and gene expression analysis (for THP-1 cell line) for IL-1β, caspase-1, and NLRP3 were performed as previously described by Davidson et al. [10].

### 2.3. Statistical Analysis

The experimental results are expressed as percentage values ± standard deviation (SD), relative to a negative control group (untreated cells). Statistical analyses were conducted using GraphPad Prism 8.0.1 (GraphPad Prism^®^, 2018; San Diego, CA, USA), employing a one-way ANOVA followed by Tukey’s test, and a two-way ANOVA followed by Tukey’s test for thermal stability evaluation of NC-ECs and the safety profile of NC-ECs and NC-Bs. A *p*-value < 0.05 was considered to be statistically significant. For gene expression analysis, calculations were performed using the 2^−ΔΔCt^ method, and results were normalized against the ACTB housekeeping gene. The results are expressed as the fold change compared to the control group.

## 3. Results

### 3.1. Part I

#### 3.1.1. Physicochemical Characteristics of NC-ECs and NC-Bs

The prepared NC-ECs and NC-Bs had average sizes of 164 (Appendix A) and 156 nm (Appendix A), respectively. NC-ECs refers to nanocapsule formulations containing epicatechin, while NC-Bs refers to nanocapsules formulations without the addition epicatechin. Both formulations had an average PDI of 0.150 to 0.160, consistent with a homogeneous particle size. In addition, both had a negative zeta potential and acidic pH, consistent with the formulation constituents. The NC-ECs were found to contain 100% epicatechin with a 96.15% encapsulation efficiency (Appendix A), and using atomic force microscopy NC-ECs and NC-Bs exhibited a droplet-like morphology with sizes of approximately 200 and 300 nm, respectively (Table 1).

#### 3.1.2. NC-EC Stability by the Centrifugation Test

A centrifugation test was performed to check for any instabilities in the prepared NC-ECs, before and after samples were conditioned for the test. As can be seen in Figure 2, no macroscopic visual changes were detected after centrifugation.

#### 3.1.3. NC-EC Thermal Stability

The stability of the prepared NC-ECs was evaluated by monitoring a set of physicochemical parameters for 45 days after formulation preparation, under three different temperature conditions (room temperature, refrigeration, and a climate chamber) (Figure 3). At 60 days after preparation, the samples presented with a foul odor and the growth of colonies of microorganism was visible at the bottom of the vials. Therefore, the analysis was discontinued after this period (Appendix A).

When stored at room temperature or in the climate chamber, the average particle size was observed to increase to approximately 200 nm, which was statistically significant. However, no significant change in particle size was observed for samples stored under refrigeration (Figure 3A). A statistically significant increase in PDI was measured for nanocapsules stored at room temperature (from 0.111 ± 0.01 at t = 0 to 0.244 ± 0.006 at t = 45 days at room temperature), while for NC-ECs stored under refrigeration or in a climate chamber, no significant changes were seen (Figure 3B). However, despite the change in this parameter, particle size distribution is still considered to be homogeneous up to 0.3. Zeta potential values also decreased significantly to around −20 mV in NC-ECs stored at either room temperature or in a climate chamber, while no significant change was observed for samples stored under refrigeration (Figure 3C). Measured pH values for the NC-ECs remained consistently low, both following preparation and throughout the 45 days of analysis. However, a small increase in pH was measured under the three temperature conditions, which may indicate early degradation of epicatechin (Figure 3D).

### 3.2. Part II

#### 3.2.1. Protein Corona Effect by NC-ECs

At concentrations of 0.01 and 0.1 μg/mL, at baseline (t = 0 h) increases in particle size (Figure 4A), PDI (Figure 4B), and zeta potential (Figure 4C) were observed, along with a significant increase in pH (Figure 4D). These results were significant based on comparison with the characterization of pure NC-ECs, without contact with the culture media, indicating the formation of a protein corona. In contrast, NC-ECs at a concentration of 50 μg/mL did not show significant changes in any of the physicochemical parameters evaluated in both culture media and incubation periods tested, indicating greater stability compared to the other concentrations evaluated.

#### 3.2.2. In Vitro Safety Profile of NC-ECs and NC-Bs

Both NC-ECs and NC-Bs had comparable safety profiles, in terms of cell viability, NO, ROS, and dsDNA levels at the different concentrations and incubation times tested (Figure 5A–D). In general, no statistically significant changes were observed at lower concentrations when compared to the negative control, suggesting that both nanocapsules have a favorable biocompatibility profile. Cytotoxic effects were consistently detected only at the highest concentrations (10 and 50 µg/mL), regardless of the incubation period. A proliferative effect was observed at intermediate concentrations (0.05 and 0.1 µg/mL), particularly after 48 and 72 h of exposure. The overall similarity between NC-ECs and NC-Bs suggests that the observed cytotoxicity was primarily related to the nanocapsule structure itself, and not to the presence of epicatechin. The statistical differences are provided in Appendix A, where the same results are demonstrated in a column graph to facilitate their visualization.

#### 3.2.3. Anti-Inflammatory Effects of NC-ECs via NLRP3 Inflammasome Modulation in Monocytes and THP-1-Derived Macrophages

The anti-inflammatory effects of NC-ECs were evaluated by monitoring NC-EC-mediated modulation of the NLRP3 inflammasome. For this assay, THP-1-derived monocytes and macrophages were activated by LPS + nigericin and treated with increasing concentrations of NC-ECs (0.01–50 μg/mL). Images obtained via optical microscopy after cell activation with LPS + nigericin and treatment with MCC950, as well as the effect of increasing concentrations of NC-ECs are shown in Figure 6A (macrophages) and 6F (monocytes).

Macrophages activated by LPS + nigericin showed reduced cell viability (Figure 6B) and increased NO levels (Figure 6C), as well as increased ROS (Figure 6D) and extracellular dsDNA (Figure 6E) when compared to untreated control cells. In contrast, cells treated with MCC950, especially at lower concentrations of NC-ECs (0.01 to 0.1 μg/mL), were able to reverse inflammatory activation caused by LPS + nigericin, compared to the positive control, suggesting that such concentrations of NC-ECs are effective in modulating the NLRP3 inflammasome. Treatment with NC-ECs was associated with a consistent anti-inflammatory effect comparable to that observed in macrophages, with greater activity observed at lower NC-EC concentrations (Figure 6G–J). Notably, the lowest concentration tested (0.01 µg/mL) was able to modulate the NLRP3 inflammasome in monocytes activated by LPS + nigericin, resulting in a significant reduction in the gene expression of NLRP3 (Figure 6K), IL-1β (Figure 6L), and caspase-1 (Figure 6M).

## 4. Discussion

In the present study, NC-ECs were developed and characterized in terms of their physicochemical parameters, in vitro safety profile, and in vitro anti-inflammatory activity.

Nanoencapsulation has the potential to overcome some of the physicochemical limitations of epicatechin, resulting in enhanced biological effects while protecting the molecule from environmental and biological degradation. Although bioactive molecules have significant potential in therapeutic applications, the clinical applications of flavonoids, and in particular epicatechin, have been limited by their chemical instability, which is due to the presence of free hydroxyl groups [30]. In addition flavonoids often have low bioavailability, because dietary polyphenols are predominantly present in a glycosylated form with one or more sugar residues conjugated to a hydroxyl group and/or aromatic ring, resulting in low intestinal absorption, with only a small amount of absorption occurring (5 to 10%) mainly in the colon [31,32]. Moreover, epicatechin is sensitive to environmental and biological enzymatic degradation, mainly due to the effects of stomach acids [33]. Epicatechin has also been shown to be susceptible to thermal degradation, with losses of 62.5% after approximately 10 min exposure to boiling water [12], as well as instability associated with photoinduced oxidation [34]. Furthermore, the solubility of epicatechin in water is hindered by its amphiphilic chemical structure, which suggests an affinity for both polar and non-polar environments: with a low polarity moiety due to the presence of a hydrophobic aromatic group, and high polarity due to the presence of hydrophilic hydroxyl groups [35]. Additionally, the LogP of epicatechin is 1.02, in agreement with the amphiphilic character described above, since LogP > 0 is an indicator of the degree of lipophilicity of molecules, where LogP < 0 is associated with hydrophobic characteristics, while LogP = 0 indicates equal solubility in both polar and non-polar solvents (DrugBank online). Because of this, the developed NC-ECs were produced by interfacial deposition using a preformed polymer method, enabling creation of a biphasic structure containing a hydrophobic core and a polymeric wall of Eudragit L-100. The NC-EC formulation was obtained with a final concentration of 0.25 mg/mL encapsulated epicatechin, and had a slightly yellowish color.

Based on measurements of the physicochemical parameters of the prepared NC-ECs, several conclusions can be made. (1) NC-EC particles had an average size of 164.7 ± 0.35 nm, confirming that they are in the nanoscale range. Because NC-Bs without epicatechin had a similar size 156.7 ± 1.69 nm, no significant change in size occurred upon the addition of epicatechin, in agreement with the microscopic findings. (2) For NC-Bs without the addition of epicatechin, PDI was found to be 0.15 ± 0.017, while for NC-ECs it was measured at 0.162 ± 0.014, indicating a homogeneous dispersion of the nanocapsule particles, with no evidence of particle aggregation [36]. (3) The zeta potential, which is an indicator of stability, was −9.06 ± 1.21 mV in NC-ECs and −14.6 ± 1.7 mV in the white NC-Bs. The negative surface charge of the nanocapsules with and without epicatechin can be explained by the presence of terminal carboxyl groups on the anionic polymer Eudragit L-100. (4) The average measured pH value for NC-ECs containing epicatechin was 3.85 ± 0.02, similar to the pH of NC-Bs without epicatechin (3.8 ± 0.02). As with the zeta potential, the acidic pH is also likely due to the presence of terminal carboxyl groups on the anionic polymer Eudragit L-100. Furthermore, data obtained here demonstrate that the content rate and encapsulation efficiency were 100% and 96%, respectively, indicating that the active ingredient epicatechin was almost completely encapsulated in the nanocarrier. These parameters found are similar to those described by Jaguezeski et al. [23], in which Eudragit L-100 nanocapsules containing 0.25 mg/mL of curcumin were prepared using the same methodology used to produce the NC-ECs, suggesting that the applied method can provide reproducible results in independent laboratories. Similarly, in a study by Cordenonsi et al. [37], Eudragit L-100 nanocapsules containing naringin and naringenin were also prepared using interfacial deposition of a preformed polymer; naringin and naringenin are also a flavonoid compounds, and data reported from that study regarding physicochemical characterization corroborate the findings of the present study.

In addition, our results are consistent with other flavonoid nanoformulations, particularly catechin-based systems. For example, poly(ethyleneglycol) (PEG)-poly(lactic-co-glycolic acid) (PLGA) nanoparticles loaded with epigallocatechin gallate (EGCG) were obtained in a size ranges of 150–200 nm, with PDI values below 0.2, and an encapsulation efficiency above 90%, with improved stability and sustained release compared to free EGCG [38]. Likewise, the chitosan–EGCG nanoparticles reported by Safer et al. [39] exhibited similar physicochemical stability with high encapsulation efficiency, reinforcing that the parameters obtained for NC-ECs fall within the expected profile for flavonoid nanoformulations.

The prepared NC-ECs were subjected to a centrifugation test to check for macroscopic signs of instability in the formulation. Visibly, after centrifugation, the nanoformulation did not show any signs of instability, such as color change or precipitation. The centrifugation test is a simple, fast, direct, and economical method for tracking any inconsistencies in the stability of nanocapsules [40]. In addition, the stability of the prepared NC-ECs was evaluated through measurement of physicochemical parameters over a 45-day period, under different temperature conditions, demonstrating that refrigeration was better for preservation of the nanocapsules’ initial characteristics immediately after preparation, similar to findings reported by Jaguezeski et al. [23], who analyzed the stability of Eudragit L-100 nanocapsules over 90 days. The statistically significant increase in average particle size observed in samples stored at room temperature and in a climate chamber suggests an early aggregation process or gradual instability over time. Furthermore, the increase in PDI in samples kept at room temperature also supports the hypothesis of instability. However, note that the values remained below 0.3, which is still considered acceptable for homogeneous nanostructured systems. In contrast, the samples stored under refrigeration maintained a more stable particle size, PDI, and zeta potential, indicating that refrigeration is more suitable for preserving the physicochemical integrity of the NC-ECs. The acidic pH of the formulation was maintained throughout the 45-day period, with slight increases under all tested conditions. This variation may indicate an initial degradation process involving epicatechin, possibly associated with oxidation [41,42]. The appearance of a foul odor and microbial growth after 60 days, regardless of storage condition, indicates that the present formulation lacks an adequate preservation strategy, such as the addition of preservatives or system sterilization.

Based on the protein corona effect test, the observed changes in the analyzed indices, especially at concentrations of 0.01 and 1 μg/mL, suggest the formation of a protein corona effect. However, both culture media have several proteins, carbohydrate molecules, and other nutrients that provide heterogeneity in the sample composition. This heterogeneity could affect the results of the test, especially with respect to the observed increase in particle size and polydispersity index parameters. Dalcin et al. [43] verified a similar corona effect formation behavior with Eudragit RS100 nanocapsules containing dihydromyricetin exposed to DMEM and RPMI culture media. The increase in surface electric charge indicates that the Eudragit L-100 polymer can interact with molecules with distinct or slightly positive charges through electrostatic attraction [44]. However, there are no reports in the literature evaluating the stability of Eudragit L-100 nanocapsules in contact with RPMI and DMEM culture media. Regarding the alkalinization of the pH after the NC-ECs came into contact with culture media, this can be explained by the fact that the culture media contains buffering components designed to maintain a neutral pH (±7.4) and establish adequate culture conditions. On the other hand, when tested at a concentration of 50 μg/mL, the NC-ECs did not show significant changes in the physicochemical parameters evaluated, indicating that there was no formation of a protein corona. It is possible that this effect did not occur because the NC-EC concentration was higher than the culture medium concentration in the dilution, compared to the other concentrations tested.

After physicochemical characterization of the prepared NC-ECs, the in vitro safety profiles of nanoparticles with and without addition of epicatechin were evaluated using a VERO cell line. VERO cells were incubated with increasing concentrations (0.01 to 50 μg/mL) of NC-ECs or NC-Bs for 24, 48, and 72 h. The results obtained show that both formulations are safe in vitro at most concentrations tested, except for the highest concentrations of 10 and 50 μg/mL. This finding suggests that the cytotoxicity is not due to encapsulation of epicatechin, but may be attributed to the formulation itself, since both NC-Bs and NC-ECs presented the same cellular toxicity profiles. Particularly through the MTT assay, it is possible to observe that the treatments do not cause a linear concentration response effect, since cells exposed to 0.01–1 μg/mL presented viability index similarly to untreated cells. This is pharmacologically acceptable considering the potential hormetic effect that can happen under treatments with bioactive molecules such as epicatechin [45]. Additionally, Siqueira et al. [46] has used a classification for cytotoxicity as follows: noncytotoxic (viability > 90%), slightly cytotoxic (viability 80–89%), moderately cytotoxic (viability 50–79%), or highly cytotoxic (viability < 50%). In this regard, most of the concentrations tested for both NC-Bs and NC-ECs are considered noncytotoxic, while 10 and 50 μg/mL proved to be highly cytotoxic. Ferreira et al. [47] also evaluated the safety profile of Eudragit L-100 nanocapsules containing naringin and naringenin in VERO cells. These nanocapsules were produced in a similar manner to the NC-ECs, and the data obtained also demonstrate that this formulation was cytotoxic, causing a significant decrease in cell viability starting at a concentration of 5 μg/mL. Although the nanocapsule constituents were added at concentrations considered to be safe [19,48,49], such an event may have occurred because monolayer cell culture is considered a simple and preliminary system for assessing the safety profile of formulations, active ingredients, and molecules, etc., in relation to more complex systems, such as assays using 3D cell culture or in vivo models [20]. An additional hypothesis for the observed cytotoxicity is based on the intrinsic characteristics of nanocapsules, due to their reduced particle size and greater potential for penetration through cell membranes [50].

To evaluate the anti-inflammatory activity of NC-ECs, macrophages were generated from human monocytes of a THP-1 cell line using PMA. To evaluate the modulation of the NLRP3 inflammasome by NC-ECs, macrophages were activated with LPS + nigericin. This methodology was previously applied by Davidson et al. [10] to evaluate the modulation of NLRP3 by the bioactive molecules catechin, apigenin, and epicatechin, both isolated and combined in macrophages derived from THP-1 monocytes. The NLRP3 inflammasome is involved in many different processes that can activate an inflammatory response, including the exposure to Pathogen-Associated Molecular Patterns (PAMPs), such as bacteria, and to Damage-Associated Molecular Patterns (DAMPs), such as ROS molecules [51]. In this regard, many research groups worldwide have been trying to investigate potential alternatives to modulate the NLRP3. To activate the NLRP3 inflammasome, first a priming signal is necessary to cause the NLRP3 gene transcription, which is followed by an activation signal, that will induce the inflammasome assembly [52]. When activated, the NLRP3 inflammasome will reflect in the release of activated caspase-1, as well as the overproduction of IL-1β and IL-18. Then, all the pro-inflammatory cytokines will be produced to overcome the inflammatory response. LPS is a molecule that is isolated from bacteria membrane. This agent can induce the NLRP3 priming stimulus [53]. Additionally, nigericin has the ability to trigger potassium efflux, being considered one of the most potent agents to probe NLRP3 assembly [54]. The results obtained showed that in general the lowest concentrations of NC-ECs were able to reverse the inflammatory activation induced by LPS + nigericin and modulate the NLRP3 inflammasome to levels similar to the inhibitor MCC950. Importantly, the lowest tested concentration of NC-ECs (0.01 µg/mL) was already effective at attenuating the inflammatory response, with consistent outcomes between macrophages and monocytes, which reinforces the robustness of this effect. Notably, we found that this same concentration was also effective in previous experiments that tested free epicatechin [10]. Combined, these results further support the potential use of epicatechin as an anti-inflammatory agent, whose mechanism of action may be via modulation of the NLRP3 inflammasome. Corroborating with these findings, and considering the described antioxidant properties of epicatechin, our data indicate that low concentrations of NC-ECs were able to decrease ROS production, which in turn may have contributed to the attenuation of NLRP3 inflammasome activation. Similarly to our observations, other studies have shown that bioactive molecules can suppress NLRP3 activation and protect against LPS-induced inflammation [53], and that modulation of the P2 × 7R/NLRP3 axis regulates macrophage responses with functional impact [55]. Together with the antioxidant properties of epicatechin, these findings support the hypothesis that NC-ECs reduce ROS and contribute to attenuation of NLRP3 inflammasome activation.

Regarding its mechanism of action, epicatechin has been widely described as having antioxidant properties [56]. Consistent with this, our findings indicate that low concentrations of NC-ECs were able to decrease ROS production, which, in turn, may have contributed to the attenuation of NLRP3 inflammasome activation. This effect is reflected in the reduced expression we observed for NLRP3, IL-1β, and caspase-1 genes in monocytes activated by LPS + nigericin. The simultaneous reduction in NLRP3 and IL-1β expression suggests that NC-ECs may interfere with the priming signal involved in inflammasome activation, while the decrease in caspase-1 expression indicates a possible impairment of the subsequent activation of the executioner component. These results corroborate the hypothesis that NC-ECs exert multi-level modulatory effects on the NLRP3 inflammasome pathway, similar to MCC950.

In summary, it was observed that both free and nanoencapsulated epicatechin are safe and demonstrate anti-inflammatory effects via modulation of the NLRP3 inflammasome in vitro, from the lowest concentrations tested. Combined, our results strongly support the potential of NC-ECs as a specific anti-inflammatory therapeutic alternative that may function via the NLRP3 pathway. Furthermore, it is important to highlight the application of nanotechnology as a tool that can protect epicatechin and direct its delivery to target cells and tissues, optimizing its biological effects. In support of this, previous studies have reported that the encapsulation of flavonoids in nanocapsules and other types of nanocarriers resulted in improved bioavailability and solubility of the tested bioactive compounds [57,58,59,60,61].

## 5. Conclusions

A nanocapsule suspension containing epicatechin (NC-EC) was obtained, presenting physicochemical parameters suitable for the methodology employed. Furthermore, lower concentrations of NC-ECs and NC-Bs did not exhibit cytotoxic effects against a VERO cell line, demonstrating good cytocompatibility. Moreover, lower concentrations of NC-ECs were able to modulate NLRP3 inflammasome activation in monocytes and macrophages activated by LPS + nigericin, demonstrating the potential anti-inflammatory effect of this formulation. However, despite the positive results reported in the present study, it is important to highlight that further pharmacokinetic and pharmacodynamic studies, as well as in vivo investigations, are still required to better understand the potential mechanisms of action of NC-ECs. Also, clinical translation pathways analyses would be interesting, in order to align the present study with the goals of nanomedicine worldwide.

## Figures and Tables

**Figure 1 biology-14-01520-f001:**
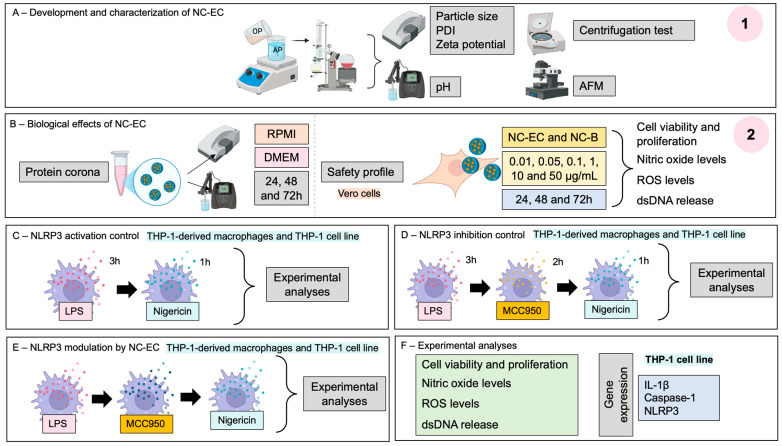
Experimental research design. (**1**) Part 1 of Materials and Methods section—(**A**) development and characterization of an NC-EC; (**2**) Part 2 of Materials and Methods section—(**B**) biological effects of the developed NC-EC; (**C**) NLRP3 activation control; (**D**) NLRP3 inhibition control; and (**E**) NLRP3 modulation by the developed NC-ECs; (**F**) Experimental analyses.

**Figure 2 biology-14-01520-f002:**
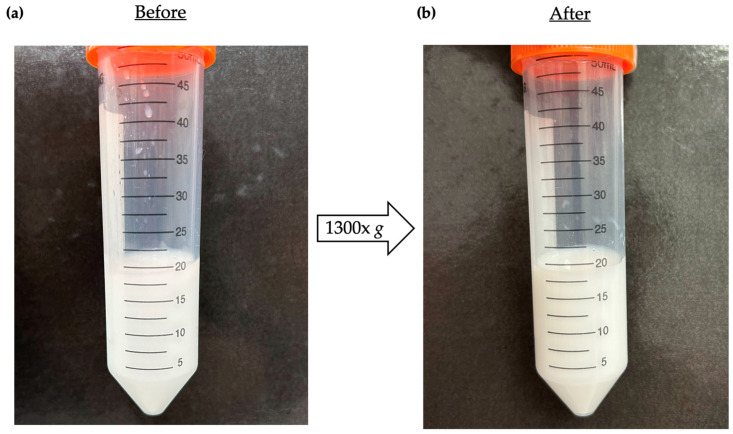
Photographs of visual macroscopic evaluation of NC-ECs before and after centrifugation. (**a**) Photograph of NC-ECs before the centrifugation test, and (**b**) after the centrifugation test.

**Figure 3 biology-14-01520-f003:**
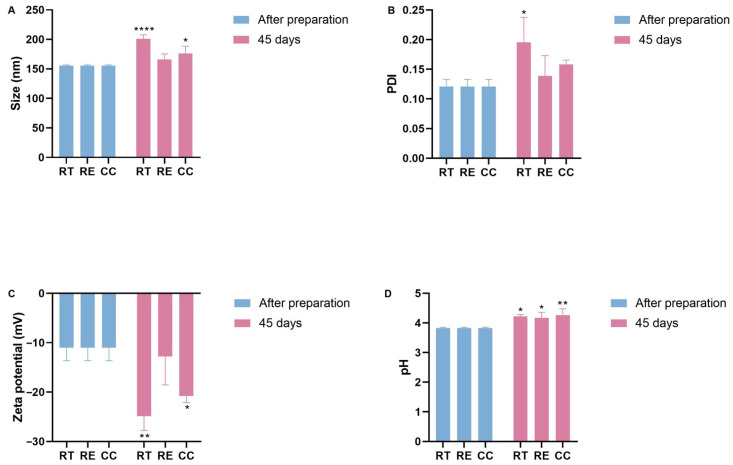
Stability of NC-ECs over 45 days under various storage conditions: room temperature (RT), refrigeration (RE), and in a climatic chamber (CC). (**A**) Average particle size (nm), (**B**) polydispersity index, (**C**) zeta potential, and (**D**) pH. Data are expressed as the mean ± standard deviation. Analyses were conducted by two-way ANOVA followed by Tukey’s test. Results with *p* < 0.05 were considered to be significant. * *p* < 0.05; ** *p* < 0.01; **** *p* < 0.0001.

**Figure 4 biology-14-01520-f004:**
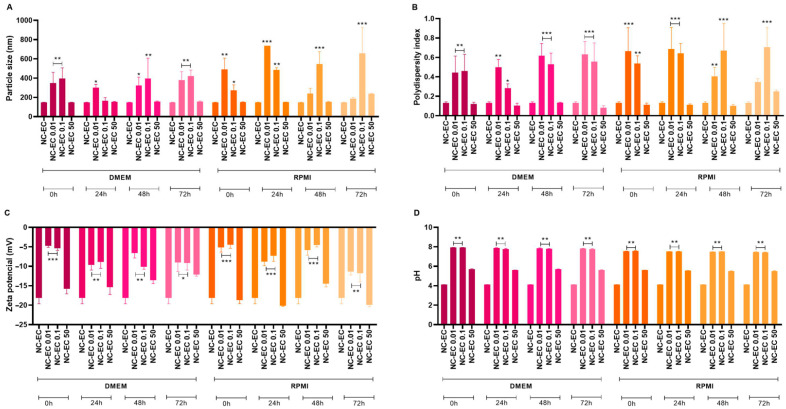
Effect of a protein corona by NC-ECs (0.01, 0.1, and 50 μg/mL) after 0, 24, 48, and 72 h of incubation in DMEM and RPMI culture media with HEPES (supplemented with antibiotics and FBS). (**A**) Particle size, (**B**) polydispersity index, (**C**) zeta potential, and (**D**) pH. Data are expressed as the mean ± standard deviation. Analyses were conducted by one-way ANOVA followed by Dunnett’s test. Results with *p* < 0.05 were considered to be significant. * *p* < 0.05; ** *p* < 0.01; *** *p* < 0.001.

**Figure 5 biology-14-01520-f005:**
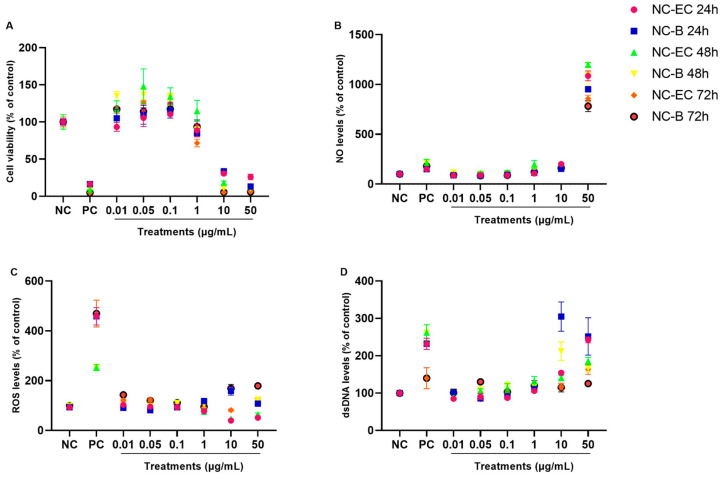
NC-EC and NC-B concentration curves—in vitro safety profile evaluation. VERO cells were exposed to different concentrations of NC-ECs for 24, 48, and 72 h. (**A**) Evaluation of predictive (24 h) and cellular regularity (48 and 72 h) indices by an MTT assay; (**B**) measurement of NO levels after 24, 48, and 72 h of incubation, respectively; (**C**) measurement of ROS levels after 24, 48, and 72 h of incubation, respectively; (**D**) quantification of extracellular dsDNA indices after 24, 48, and 72 h of incubation, respectively; NC: negative control (untreated cells); CP: cells exposed to 200 µM H_2_O_2_ for MTT, DCFH-DA and PicoGreen assays and 10 µM sodium nitroprusside for NO determination; Statistical analyses were performed by two-way ANOVA followed by Tukey’s Post Hoc test. Results with *p* < 0.05 were considered to be significant.

**Figure 6 biology-14-01520-f006:**
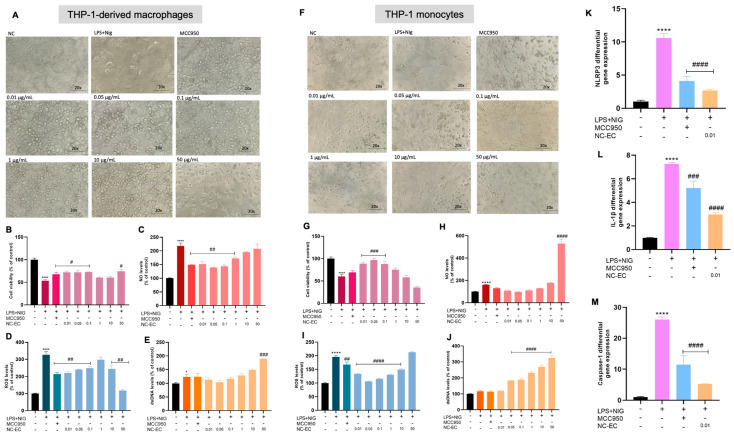
Anti-inflammatory capacity of NC-ECs via modulation of NLRP3 in macrophages and monocytes. (**A**) Analysis of macrophage cell morphology by optical microscopy after each treatment. (**B**) Macrophage cell viability after each treatment. (**C**) NO levels in macrophages after each treatment. (**D**) ROS levels in macrophages after each treatment. (**E**) Extracellular dsDNA levels in macrophages after each treatment. (**F**) Analysis of monocyte cell morphology by optical microscopy after each treatment. (**G**) Monocyte cell viability after each treatment. (**H**) NO levels in monocytes after each treatment. (**I**) ROS levels in monocytes after each treatment. (**J**) Extracellular dsDNA levels in monocytes after each treatment. (**K**) NLRP3 gene expression in monocytes after each treatment. (**L**) IL-1β gene expression in monocytes after each treatment. (**M**) Caspase-1 gene expression in monocytes after each treatment. NC: negative control (untreated cells). NC-ECs: epicatechin-loaded nanocapsules. Statistical analysis was performed by one-way Anova followed by Tukey post hoc. Results with *p* < 0.05 were considered to be significant. * Represents comparison to the negative control; # represents comparison to LPS positive control—# *p* < 0.05; ## *p* < 0.01; ### *p* < 0.001; #### *p* < 0.0001; * *p* < 0.05; **** *p* < 0.0001. Magnification: 20×.

**Table 1 biology-14-01520-t001:** Physicochemical characterization of NC-ECs and NC-Bs.

Parameters	NC-ECs [0.25 mg/mL]	NC-Bs
Size	165 ± 0.35 nm	157 ± 1.96 nm
PDI	0.162 ± 0.014	0.15 ± 0.017
Zeta potential	−9 ± 1.21 mV	−15 ± 1.7 mV
pH	3.85 ± 0.02	3.8 ± 0.02
Bioactive content	100 ± 0.40%	Not applicable.
EE	96.15 ± 1.01%	Not applicable.
AFM	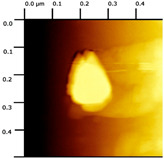	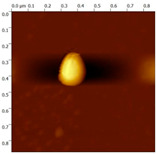

## Data Availability

Data will be available on request.

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
