# Peer review of "Epicatechin-Loaded Nanocapsules: Development, Physicochemical Characterization, and NLRP3 Inflammasome-Targeting Anti-Inflammatory Activity"

_biology, 2025, doi:10.3390/biology14111520_

Round 1
Reviewer 1 Report
Comments and Suggestions for Authors
The manuscript entitled “Epicatechin-Loaded Nanocapsules: Development, Physico-chemical Characterization, and NLRP3 Inflammasome-Target Anti-Inflammatory Activity” investigated nanocapsulation of epicatechin flavonoid to improve its applicability and evaluate whether the formulation maintains its anti-inflammatory effects via modulation of the NLRP3 inflammasome or not. This work is promising for future applications in diatery supplements.
I have the following comments:
- Add epicatechin and nanocapsule to the keyword to attract more citations.
- Add to the introduction part problem statement about inflammation.
- The authors have to explain why they use 0.025% of epicatechin not other doses.
- How was validatedthe ultrafiltration to avoid particle leakage or binding of drug–membrane.
- Explain if the used HPLC assay was fully validated (robustness, LOD/LOQ, precision, accuracy, linearity, and specificity.
- The used Corona in bioassay was inferred from pH-PDI-Size shifts. If possible, you can add the utilized method, for example LC-MS-MS.
- For biological study controls, add free epicatechin and vehicle controls at matched doses.
- Add more updated references 2024-2025. Also, the manuscript requires language polishing.
Language polishing required
Reviewer 2 Report
Comments and Suggestions for Authors
The number of papers devoted to the design of targeting drugs is growing enormously. This specifically considers targeting of natural products. Thus, the manuscript is in-line with this trend and might be published albeit several improvements has to be done before that:
1./ English has to be carefully corrected. This considers especially Experimental part. In some cases the awkward English causes difficulties in understanding the paper;
2./ there is no need to write Zeta fron capital letter, should be zeta;
3./ paragraph 2.1.1.: remive organic solven and leave only solvent (it is a water ethanol mixture);
4./ there is no need to use phrases like "Zetasizer® equipment" (remove equipment);
5./ page 5: is phase A 01.% trifluoroacetic acid?;
6./ paragraph 2.1.3.: replace "were recorded through photographs" by photographed;
7./ CO2 not CO2 (page 6);
8./last sentence in paragraph 2.2.1 is unclear;
9./ entitle paragraph 3.1.;
10./ paragraph 3.1.3.: "Regarding PDI, there was a statistically significant increase in nanocapsules" is to big shortcut;
11./ line 258: "acidic pH" is a jargon; the sa,e considers "alkanization of pH" (line 272)
12./ Quality of Figures 3-6 should be improved;
13./ line 291: what Authors mean by "proliferative effect";
14./ line 312: statement "...and treated with a NC-EC concentration curve.";
15./ Disscussion is quite chaotic;
16./ Authors please conform to the requirements of Journal when citing the literature. There are a lot of obstacles.
Reviewer 3 Report
Comments and Suggestions for Authors
This manuscript presents an innovative study on the encapsulation of epicatechin into nanocapsules using Eudragit L-100, with the aim of overcoming key limitations of flavonoid delivery. The work is significant as it not only enhances the physicochemical stability and bioavailability of epicatechin but also demonstrates preserved anti-inflammatory activity via NLRP3 inflammasome modulation. The manuscript is scientifically relevant and aligns well with the growing interest in nanotechnology-driven strategies for managing inflammation. While the study is methodologically strong, further refinements in clarity, contextualization, and translational emphasis would improve its impact and readability.
- The introduction clearly presents epicatechin’s therapeutic potential. It would benefit from a stronger rationale connecting flavonoid limitations with the choice of Eudragit L-100 nanocapsules, highlighting why this polymer is particularly advantageous.
- The preparation method is well-described, but additional justification for the selected concentration (0.25 mg/mL) and its clinical relevance would improve clarity.
- Share the compatibility study data is available.
- Physicochemical parameters are comprehensive, yet the discussion could be enriched by comparing these findings with similar flavonoid nanoformulations, situating the results within broader literature.
- The 45-day thermal stability study is valuable, but longer-term stability or accelerated stability data would enhance confidence in future clinical applicability, if available.
- Cytocompatibility in VERO cells and anti-inflammatory assays in THP-1 macrophages are appropriate. However, more detailed discussion of concentration-response effects and potential cytotoxicity thresholds would add depth.
- The role of the NLRP3 inflammasome is a strong point. Including discussion of upstream or downstream signaling pathways (e.g., caspase-1, IL-1β) could strengthen mechanistic interpretation.
- The conclusion should not only emphasize NC-EC’s potential but also highlight next steps, such as in vivo validation, pharmacokinetic profiling, and clinical translation pathways. This would give a forward-looking perspective aligned with nanomedicine goals.
- The following studies are suggested to evaluate and add to the literature review of the manuscript: https://doi.org/1007/s12012-020-09576-4, https://doi.org/10.3389/fphar.2025.1522146, https://doi.org/10.1126/sciadv.adr4894
Round 2
Reviewer 1 Report
Comments and Suggestions for Authors
The authors have successfully addressed all the required modifications, and I have no further comments
Reviewer 2 Report
Comments and Suggestions for Authors
Now paper is suitable for publication
Reviewer 3 Report
Comments and Suggestions for Authors
No more comments.